# All Norms and No Nuance Make LLMs Dull Cultural Simulators

**Saurabh Kumar Pandey, Sougata Saha, Monojit Choudhury**
Mohamed bin Zayed University of Artificial Intelligence,
{saurabh.pandey, sougata.saha, monojit.choudhury}@mbzuai.ac.ae

## Abstract

Socio-demographic prompting (SDP), which prompts Large Language Models (LLMs) to generate culturally aligned behaviors using demographic proxies, is commonly used to assess cultural biases in LLMs. However, its sensitivity to the prompt raises questions about its reliability in cultural assessment and user behavior simulation. Here, we explore inverse socio-demographic prompting (ISDP), a method that prompts LLMs to predict users' cultural backgrounds based on their behaviors, offering a robust alternative by mapping behaviors to cultural proxies. We evaluate SDP and ISDP across four LLMs - Aya-23, Gemma-2, GPT-4o, and LLama-3.1 - using the Goodreads-CSI dataset (Saha et al., 2025), which captures cross-cultural non-understandability in book reviews from users in India, Mexico, and the USA. Our analysis reveals that ISDP is a much more robust way of assessing LLMs' cultural alignment than SDP. Next, we simulate user behavior and evaluate model performance by aggregating behavior at different levels. We observe that at a group-level, GPT-4o excels in ISDP with actual user behavior and struggles when the behavior is LLM-generated. Furthermore, at the user level, GPT-4o performs best when the behavior is generated by itself or by the actual users. In contrast, at the group level, other models perform better with LLM-generated behavior than with the actual user behavior. We reason that this is likely because LLMs generate stereotypical outputs due to maximum likelihood decoding, which deviates from real-world user behavior, which is more nuanced and less normative - individuals do not exhibit all stereotypes of a culture. These findings have significant implications for simulating user behavior using LLMs and position ISDP as a valuable framework for understanding the limitations of user behavior simulation and studying cultural representation in LLMs.

## 1 Introduction

Studies on Large language models (LLMs) have consistently shown high cultural biases in their generated responses concerning the representation of various demographic groups (Bender et al., 2021) and cultures (Masoud et al., 2023). Recent studies have focused on probing models for assessing cultural knowledge and biases using culturally conditioned prompts commonly referred to as *socio-demographic prompting* (SDP) (Li et al., 2024b; AlKhamissi et al., 2024; Wan et al., 2023), or testing for specific sociocultural knowledge through specially curated datasets, such as (Nguyen et al., 2023; Dwivedi et al., 2023; Fung et al., 2024). However, SDP has been shown to be sensitive to minor variations in the prompt and, therefore, could lead to potentially misleading results (Mukherjee et al., 2024; Beck et al., 2024).

SDP implicitly assumes that demographic groups exhibit stereotypical behaviors that models should be able to reproduce (Naous et al., 2023; Kotek et al., 2023; Shrawgi et al., 2024). Knowledge-retrieval-based approaches also assume that certain knowledge is central to and therefore known to all or most members of a group, which the models must know as well (Nguyen et al., 2023; 2024; Shen et al., 2024). Although there is practical utility

in such knowledge of stereotypical behaviors, any group of users, loosely defined by a demographic proxy (Adilazuarda et al., 2024) such as country or religion, or a combination of these, exhibits a wide range of *behaviors*. We define "behavior" broadly as the preference for certain values or artifacts, beliefs in certain facts, etc. (Hogg, 2016). We argue that cultural awareness in a model encompasses as much understanding and knowledge of this variation of user behavior as it is about the knowledge of the average or stereotypical behavior. This broader form of cultural awareness can be tested by reversing the SDP task, that is to say, by providing the user behavior and asking the model to guess the probable membership of the user across different demographic groups. We term this technique as *Inverse Socio-Demographic Prompting* or ISDP, for short.

The following Bayesian argument provides a simple justification in favor of our claim. Imagine that a demographic group $g$ is described through certain *demographic proxies* (Adilazuarda et al., 2024). A user $u$ is sampled from this group, and we observe their behavior $b$. This situation can be represented by the classic *noisy channel model* (Jurafsky & Martin, 2016), where the channel model, $P(b|u)$ can be further approximated by $P(b|g)$, which is equivalent to saying that $b$ is independent of $u$ given $g$. SDP asks the model to produce $b$ given $g$, or in other words, to estimate $P(b|g)$. However, because the space of behaviors, $b$, is large and complex, the dependence on $u$ cannot be completely ignored (i.e., the independence approximation does not hold); typically, one would need to sample a large number of points from the model as well as the real world to compute and compare the predicted and actual distributions. On the other hand, the inverse task (ISDP), asks for $g$, given the behavior $b$, which can be used to estimate $P(g|b)$ for a model. By Bayes' rule,

$$P(g|b) = \frac{P(b|g)\ P(g)}{P(b)}$$

Since the space of $g$ and the entropy of $P(g)$ are expected to be much lower than that of $P(b)$, the entropy of $P(g|b)$ would be (much) lower than that of $P(b|g)$ which would make it much easier to obtain reliable estimates of $P(g|b)$ than $P(b|g)$ from the real world and the model. This asymmetry is due to the fact that *no member of a coherent socio-cultural group displays all the prototypical behaviors of a group. However, all members show at least some prototypical behavior of the group* (Leung & Cohen, 2011; Morris et al., 2015; Hogg, 2016).

In this paper, we present a set of studies that illustrate the efficacy of ISDP in estimating cultural knowledge of LLMs. We conduct our experiments on the Goodreads-CSI dataset recently introduced by Saha et al. (2025), which is a user-annotated dataset of culture-specific items (CSIs) (Aixelá, 1996) from Goodreads reviews that users from 3 countries (USA, Mexico, and India) found difficult to understand[1]. Two features that make this dataset interesting and challenging are: (a) As illustrated by the authors, the inter-annotator agreement between the non-understandable items (CSIs) marked by users from the same country is low, which is also observed between the users and the LLMs, alluding to the high entropy of $P(b|g)$. (b) The task involves estimating "what a user does not know" rather than the more usual task of predicting "what a user typically knows", making the space of possible behaviors even larger and more complex.

On this dataset, we aim to test the following two hypotheses: (**H1**) Predicting the group, $g$, from the user behavior, $b$, leads to more reliable measurements of a model's cultural knowledge and biases. (**H2**) Given two models $M_1$ and $M_2$, if $M_1$ is tasked to simulate the behavior of a user from a group $g$ (in this case, country) using SDP, and $M_2$ is asked to guess $g$ from this simulated behavior $b_{sim}$ using ISDP, models will have a higher agreement between themselves than when asked to predict $g$ from a real user's behavior $b$, even when the behaviors $b$ and $b_{sim}$ are aggregated over reviews, or users from the same location. These hypotheses arise from the observation that models trained on similar datasets are likely to generalize in comparable ways and will have similar biases.

We conducted experiments with four LLMs - Aya, Gemma, GPT-4o, and Llama. The inter-model agreement is much higher in the case of ISDP, validating **H1**, but to our surprise, **H2** turned out to be false. When behaviors of multiple users are aggregated, GPT-4o is better at predicting $g$ based on actual user's $b$ than that from the simulated personas by

---

[1]Note: In this dataset, a user behavior is the action of highlighting a span as difficult to understand.

other models or even itself, implying that for the task studied, models possibly generate stereotypes and cannot consistently and reliably simulate the behavior of the real users, irrespective of the demographic group they belong to. In contrast, other models are better at predicting $g$ based on aggregated simulated user behavior $b_{sim}$ than $b$, indicating that these LLMs are aware of the cultural stereotypes more than actual user behavior, indicating strong biases and their unsuitability for use across diverse cultural contexts. Interestingly, the trends are reversed at the individual user level, where GPT-4o predicts $g$ better when $b_{sim}$ is generated by itself rather than actual $b$. Other models falter with GPT-4o-generated behavior at an individual level, indicating that GPT-4o possibly generates the least stereotypical behavior for $g$ compared to other LLMs. Our findings highlight critical challenges for social simulations using LLMs, indicating that they can only simulate user behavior to an extent, beyond which they become stereotypical. However, GPT-4o shows promise in this domain, outperforming other LLMs in associating nuanced, non-stereotypical behaviors with cultural contexts, particularly at the individual level, making it a more suitable choice for such simulations.

The rest of the paper is organized as follows. In Section 2, we present the materials and methods of our experiment. Section 3 presents the findings. We discuss, raise open questions, and conclude in Section 4, while highlighting related works.

## 2 Methodology

Figure 1 illustrates our study's setup comprising two phases: Phase 1- socio-demographic prompting (SDP) and Phase 2- inverse socio-demographic prompting (ISDP). In this section, we present the detailed methodology and the experimental setup for both phases.

### 2.1 Socio-Demographic Prompting (SDP)

Socio-demographic prompting encompasses probing models with prompts containing cultural (demographic) proxies[2] to generate culturally aligned behavior as response. By simulating user personas on the Goodreads-CSI dataset, Saha et al. (2025) evaluated GPT-4o as a cultural reading assistant through socio-demographic prompting. The users were defined at the intersection of two proxies - *country* and *genre preference*. Using the prompt in Appendix A.1, we prompt Aya-23, Gemma-2, and Llama-3.1 to identify CSIs in Goodreads reviews by simulating user personas. Since users can mark any contiguous span of text as a CSI, the dataset had a high degree of variability in the annotations. This variability may lead to a mismatch between spans identified by users and the models. To alleviate these issues, we collected all the CSI spans annotated by users and models for a given review text and manually standardized the spans for each review text. In this work, we used this standardized Goodreads-CSI dataset for all subsequent experiments. The details of the standardization process are present in Appendix B.

### 2.2 Inverse Socio-Demographic Prompting (ISDP)

To measure the performance of LLMs in predicting cultural background from user behavior, we utilize ISDP, outlined as Phase 2 in Figure 1. Here, we reverse the task of SDP and instead prompt models to predict cultural proxies from user behavior. For the Goodreads-CSI dataset, we prompt models to predict the country as the cultural proxy $g$, given the review text and the CSI spans annotated by actual users $b$ and user-simulated personas in SDP $b_{sim}$. Specifically, given a set of behavior, we prompt the models to rank three countries, India, Mexico, and the USA, according to how likely is the behavior from an individual or a group of users from each country.

$$Rank(\{P(g = c|s)|c \in \{India, Mexico, USA\}\})|s \in \{b, b_{sim}\}$$

The country with rank 1 indicates individuals or groups from that country would most likely not understand the CSI spans ($b, b_{sim}$). The country ranked 3 is where the CSIs are least likely to be not understood.

---

[2]Please refer to Adilazuarda et al. (2024) for detailed definition of cultural proxies.

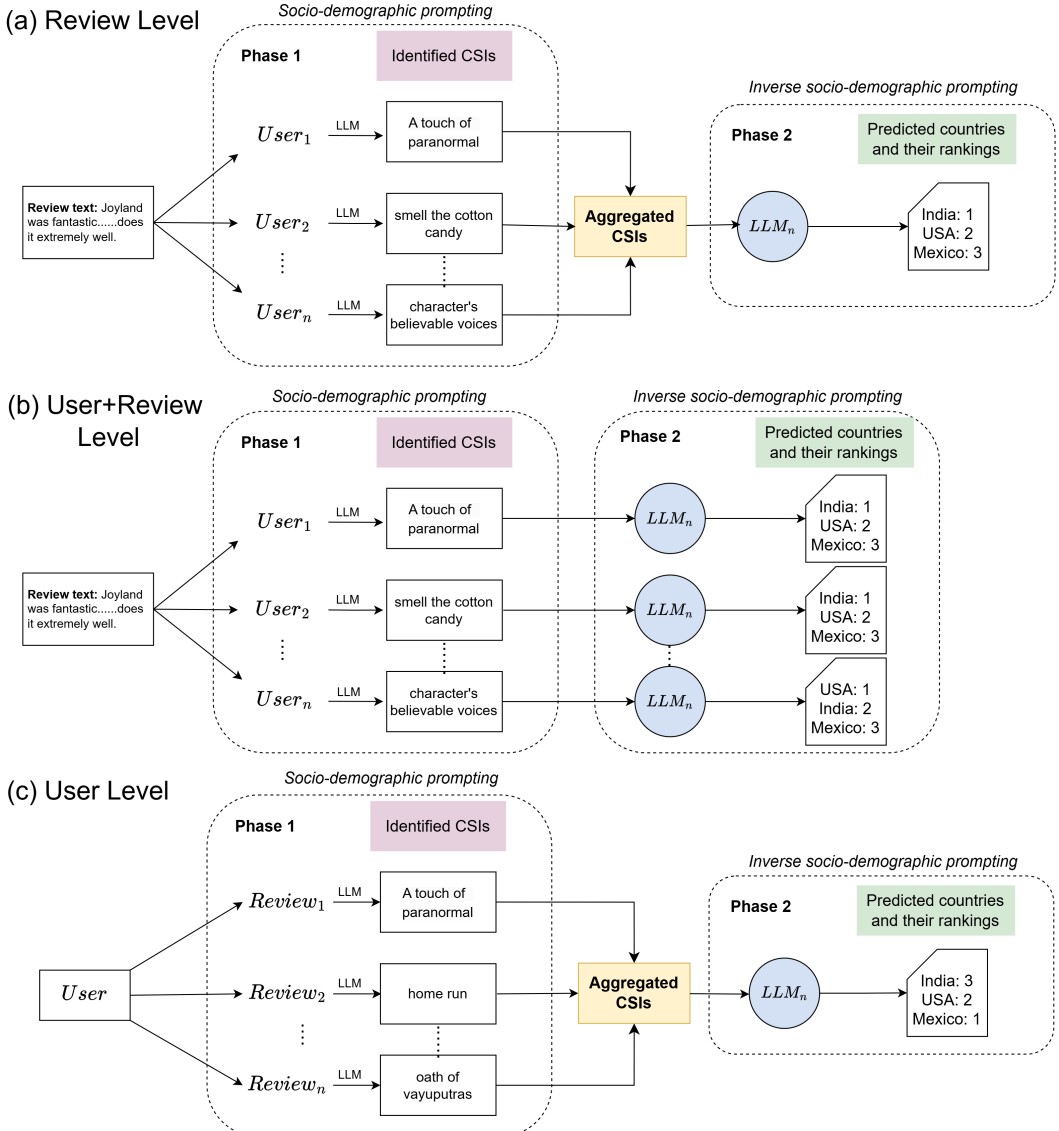

Figure 1: Flow diagram of our two-step process for three experimental setups. In Phase 1 (SDP), we prompt different LLMs and ask users to identify CSIs in a given review text. In Phase 2 (ISDP), we aggregate CSIs at different levels and prompt an $LLM_n$ under test to predict the cultural background (country). We prompt other LLMs similarly under three different aggregation schemes - (a) Review Level, (b) User+Review Level, and (c) User Level, as explained in Section 2.3.

## 2.3 Dataset

We use the Goodreads-CSI dataset (Saha et al., 2025) which contains user annotations of 57 English reviews of books originating from India, UAE, and the USA. The annotations are performed by 50 users, where 8 are from India, 22 are from Mexico, and 20 are from the USA. Given a book review, each user identifies text spans of varying lengths that they found difficult to understand, which are termed as culture-specific item (CSIs). Additionally, to test our second hypothesis (**H2**), we prompt LLMs in a SDP framework to generate model-simulated user behavior at a country and genre preference level for each of the 57 review texts across the 3 countries, as detailed previously in Section 2.1. We also aggregate the behavior (CSI spans) at three different levels - review level, user + review level, and user

level and perform ISDP to measure models capacity in handling varying types of behavior. To neutralize the effect of any positional bias in the prompt options, we construct prompts with all possible permutations of country ordering as the options. The prompt statistics are detailed in Table 1, and the actual prompts for all experimental setups are presented in Appendix A.2.

| Aggregation Level | Models | | | | | Average Input Prompt Tokens (#) |
|---|---|---|---|---|---|---|
| | Aya-23 | Gemma-2 | GPT-4o | Llama-3.1 | User | |
| Review | 180 | 180 | 180 | 180 | 167 | 501.06 |
| User + Review | 360 | 358 | 348 | 360 | 511 | 1553.73 |
| User | 48 | 48 | 46 | 48 | 90 | 499.00 |

Table 1: Prompts statistics for each level of behavior aggregation, across models and user.

**Review level:** Given a review text, we obtain a distribution of behaviors by aggregating the CSI spans identified by all the users who annotated the review text from a country. We filter out scenarios where a set of behavior maps to multiple counties and only preserve the cases with 1:1 behavior-country mapping, which amounts to 3,800 prompts (25% of total possible prompts). We prompt models with the review text and the aggregated behavior and task them to rank the countries based on the likelihood of a country exhibiting the behavior distribution. Thus, enabling us to assess the discriminative capability of the models from behavior and their ability to map user behavior to culture. This is the highest level of behavior aggregation in our setup.

**User + Review level:** Unlike review level we do not aggregate the user behavior for reviews and instead prompt models with each individual behavior and the review text and task it to generate a ranking over the countries. Similar to review level, filtering non unique behavior-country mapping results in 5,500 (17% of total possible prompts).

**User level:** For the user level, we formulate a prompt with all the (review, CSIs) pairs that a user has annotated. Since a user may annotate many reviews, we generate multiple prompts for a single user to keep the prompt length under 2K tokens. Since the member of any socio-cultural group usually exhibit only some of the prototypical group-level behaviors, this setup helps us measure the effectiveness of models in predicting cultural background from individual preferences.

### 2.4 Models

We experiment with open-weights models such as `Aya-23-8B` (Aryabumi et al., 2024), `Gemma-2-9B-it` (Team et al., 2024), and `Llama-3.1-8B-Instruct` (Dubey et al., 2024), and the closed-source `gpt-4o-2024-05-01-preview` (Achiam et al., 2023). We prompt the open-weights models to get the long-form generation and post-process the generated response in a desired JSON format using GPT-4o. For GPT-4o, we directly get the outputs in the desired JSON format. Posing ISDP as a ranking problem (as discussed in Section 2.2), we calculate the Mean Reciprocal Rank (MRR) to evaluate model performance. We use the paired t-test to assess the statistical significance of all our results, which are presented in Tables 2, 3, and 4 in Appendix A.3.

## 3 Results and Analysis

### 3.1 Effect of Prompt Ordering

Figure 2 shows the effect of the ordering of the prompt options on a model's response, at each level of behavior aggregation. We measure *Response Consistency* (plotted on the primary Y-axis) as the the proportion of time the model predicts the same country in the first position (rank=1), to the same behavior, irrespective of which country appeared as the

first option in the prompt. We observe GPT-4o's responses are most consistent, and is most likely invariant to the option ordering. The responses of Aya and Llama are less consistent, indicating possible positional bias.

For each option ordering, we also check if the top predicted country (rank=1) matches the first option provided in the prompt and compare the distribution across ordering pairs. If the top predicted country is affected by the first option then there should be significant difference in the distributions across ordering pairs. We perform a paired t-test on the distributions from all possible pairs of option orderings for a model, with the null hypothesis: *The distribution of whether the top predicted country matches the first option between two orderings are the same.* The alternate hypothesis being the distribution changes across orderings, which might indicate influence of the option ordering in the prompt.

The secondary axis (Num Significant Pairs) plots the proportion of times the t-test results are significant with a $p < 0.05$. We observe that the null hypothesis cannot be rejected for GPT-4o in Review and User+Review Levels. This indicates that GPT-4o is least affected by the option ordering at Review and User+Review levels. Compared to GPT-4o and other models, we observe that the null hypothesis can be rejected for most of the cases for Aya at User level, indicating that Aya's responses are more random at the User level and possibly sensitive to the other aspects of the prompt.

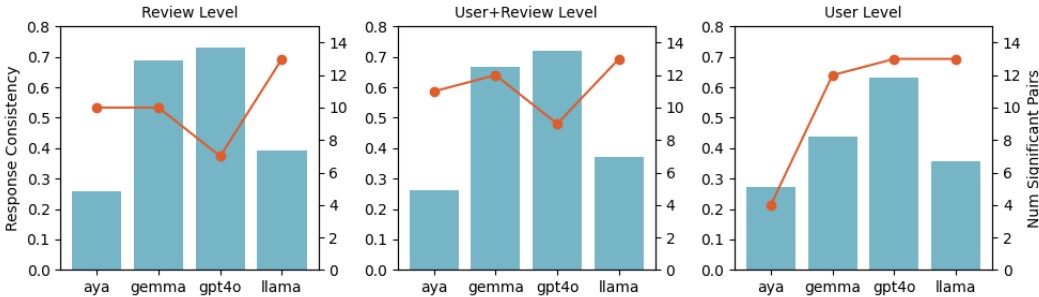

Figure 2: Model-wise effect of prompt option ordering at each level of behavior aggregation.

## 3.2 LLMs as Generators (Socio-demographic prompting)

For SDP, we calculate Inter Annotator Agreement (IAA) using Krippendorff's alpha among models, users, and user-model. As depicted in Figure 3, we observe that the user-user IAA and the model-model IAA are significantly higher than the user-model IAA, implying that models and users agree more among themselves on what is a CSI and exhibit similar behaviors among themselves rather than with each other. The model-model IAA values are also higher in comparison to the user-user IAA, which indicates that the models show similar typical or stereotypical behavior. Users, on the other hand, exhibit varied behavior and hence have lower IAA scores.

## 3.3 LLMs as Discriminators (Inverse socio-demographic prompting)

Figure 4 plots the MRR of the models at all three levels of behavior aggregation. The spread of the boxes depicts the MRR variation due to change of ordering of the countries in the prompt's options. The red dotted line ($Y = 0.61$) depicts the random baseline, considering each option is likely with 1/3 probability. We observe the following.

**Review and User+Review Levels:** As depicted in Figure 4 *Review level*, for user spans, GPT-4o has the highest average MRR. Although, the scores are not statistically different ($p$-value $< 0.05$) across models. Also, irrespective of the level of behavior aggregation, the average MRR of almost all models are greater than the baseline for user-generated spans. Comparing this with the IAA scores between user and model generated behaviors

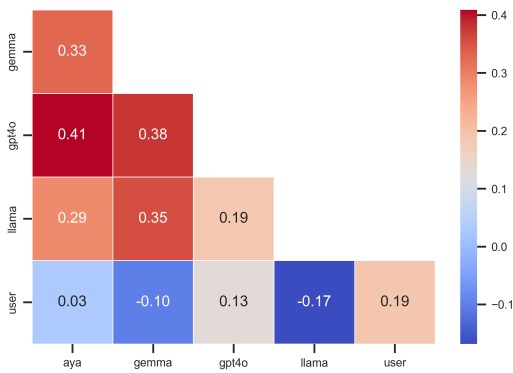

Figure 3: IAA between models and humans for SDP.

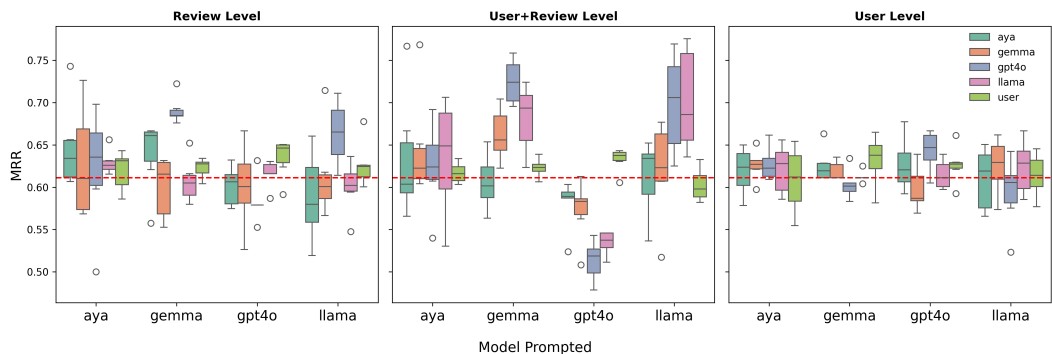

Figure 4: Model wise MRR scores across five different sources of behavior (user and LLM-generated) and three levels of behavior aggregation.

for SDP clearly indicates that models are better at discriminating between groups $g$ given a behavior $b$ (ISDP), rather than generating a behavior for a group (SDP). Thus, proving our first hypothesis (**H1** in Section 1) correct.

Interestingly, for *Review* and *User+Review* levels of, GPT-4o performs worse when the spans are generated by itself or other LLMs $b_{sim}$, compared to user-generated spans $b$. Given that users generally exhibit only a few prototypical behaviors (and not all) of a culture, and since GPT-4o better aligns with users both for SDP and ISDP, we hypothesize that it is better at discriminating stereotpical behavior from non-stereotypes compared to other models. As discussed previously, since LLMs usually perform worse in SDP than ISDP, it is likely that GPT-4o (and other LLMs) usually generate stereotypes during SDP. However, since GPT-4o is more aligned with users, it fails to map model-generated stereotypes $b_{sim}$ to a country $g$ which other models do better. This raises question on the nature of data that other LLMs are trained on, which likely promotes associating stereotypes with groups. Among the other LLMs, Gemma performs best for the GPT-4o-generated spans, followed by Llama and Aya.

Except for Llama-3.1, the MRR scores for all models are less varied for the user-generated spans at *User+Review* level than the *Review level*. Furthermore, the average MRR scores are amplified for the GPT-4o-generated behaviors, compared to the *Review level*.

**User Level:** The average MRR scores across models are less varied and and their performance is near baseline. Intersetingly, GPT-4o performs best on spans generated by itself, which sharply contrasts the trend observed at *Review* and *User+Review* levels. We hypothesize that since all the behaviors for a user across all reviews are presented together at the *User* level, the aggregated behavior better reflect the non-stereotypes, contrary to the *Review* and *User+Review* levels. Thus, further consolidating our previous hypothesis that GPT-4o is

a better discriminator of non-stereotypical behavior, contrary to our hypothesis **H2** from Section 1.

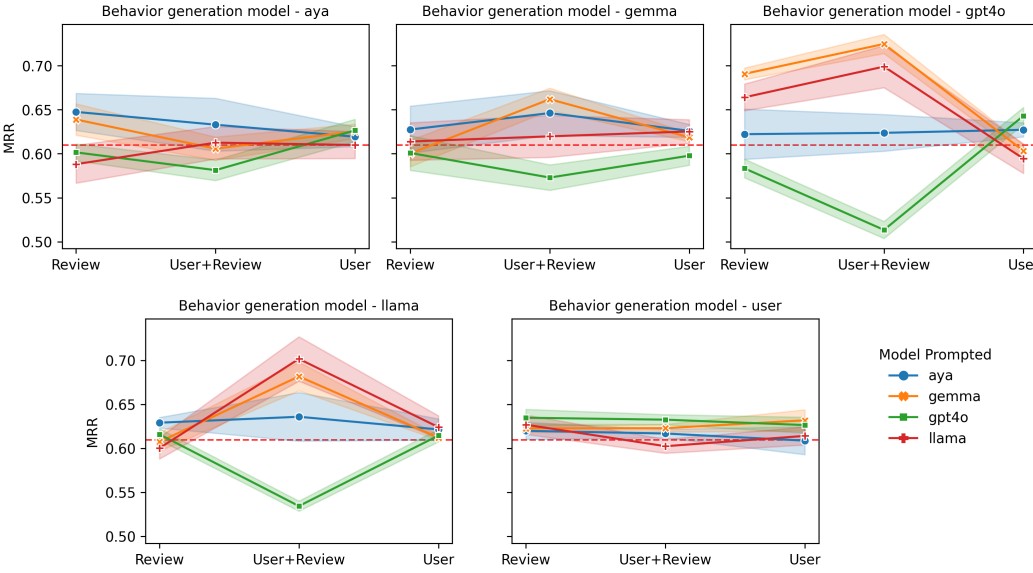

Figure 5: Average MRR scores for all combinations of generators and discriminators.

Furthermore, we plot the average MRR scores between different pairs of behavior generators and discriminators in Figure 5 and observe hypothesis **H2** from Section 1 to not hold true as there are cases, apart from GPT-4o, where models are better discriminators when the behavior is generated by users rather than LLMs.

Figure 6: Average MRR scores for different levels of behavior.

## 4  Discussion and Conclusion

**Related Work:**   Predicting and modeling human behavior has always been a challenging task (Cui et al., 2016a;b; Wang et al., 2024a). Measurement of cultural awareness in LLMs inherently requires modeling and/or prediction of human behavior, which is typically conducted using SDP on specifically curated datasets under culture-specific settings (Nguyen et al., 2023; Dwivedi et al., 2023; Fung et al., 2024; Shi et al., 2024; Nadeem et al., 2021; Wan et al., 2023; Jha et al., 2023; Li et al., 2024b; Cao et al., 2023; Tanmay et al., 2023; Rao et al., 2023; Kovač et al., 2023). Studies have also evaluated LLMs' knowledge of cultural artifacts such as food, art forms, clothing, and geographical markers (Seth et al., 2024; Li et al., 2024a; Koto et al., 2024). However, many of these methods argue that there is a need for development of robust evaluation benchmarks that can test the cultural understanding in LLMs (Wang et al., 2024b; Rao et al., 2024; Myung et al., 2024; Zhou et al., 2024; Putri et al.,

2024; Mostafazadeh Davani et al., 2024; Wibowo et al., 2024; Owen et al., 2024; Chiu et al., 2024; Liu et al., 2024; Koto et al., 2024). The Goodreads-CSI dataset, recently introduced by Saha et al. (2025), serves as a robust benchmark as they capture CSIs- a term introduced by Aixelá (1996) and further explored in various works (Pandey et al., 2025; Zhang et al., 2024; Daghoughi & Hashemian, 2016; Narváez & Zambrana, 2014; Sperber et al., 1994; Trivedi, 2008), which depicts things that people would not understand due to their culture.

**Open Questions and Conclusion:**   Treating CSI identification - identifying what people will not understand - as the behavior, in this work we evaluated ISDP as a promising and robust paradigm to test if LLMs can map behavior to country (proxy for culture), which indicates their cultural suitability. The three behavior levels, *Review*, *User+Review*, and *User*, depicts behavior on a continuum from a general (overall) level to a user. We plot the trend of the average MRR score across the three levels of behavior aggregation for each model in Figure 6.

We observe that for all models, except GPT-4o, the MRR scores are consistently low (closer to the random baseline) across all levels for the user-generated behavior, whereas the scores are higher for LLM-generated behavior at *Review* and *User+Review* levels, especially when the generators are GPT-4o and Llama. In contrast, although GPT-4o's scores are consistently low (similar to other models), its performance trend on LLM-generated behavior is opposite, as discussed previously. It performs low at a *Review* level, which further degrades at a *User+Review* level, but outperforms all models at the *User* level, flipping the trend. We observe a similar trend for behaviors generated by other LLMs, more evident in Llama than in Gemma and Aya. These trends indicate that GPT-4o is possibly better at personalization to users, and possibly models user behavior better than other models. Furthermore, this also indicates that other models are essentially learning stereotypes, which further raises questions on the nature of their training data. This might also indicate that the other (smaller) models are trained on GPT or other LLM-generated data, causing them to model user behavior differently than evident in the real world, ascertaining which we leave as future work.

**Repercussions:** Since LLMs are being used in several everyday applications, strong alignment of their generated behavior to over-represented norms or *typical* behavior poses serious risks, not only for the underrepresented communities (say, Mexico or India), as the narrative so far has emphasized, but also for users from over-represented groups (say, USA). This is because, as our study shows, every user exhibits some behavior which is not part of the norm. This was demonstrated by Agarwal et al. (2024) in their study on writing style, where they showed that LLM-assisted writing results in convergence of styles for users from both USA and India. However, the degree of loss of style diversity was greater for users from India (a further underrepresented group) than for those from USA. Our study sheds light on why this might be the case, not only for writing styles but for any other aspect of user-behavior that LLM-driven applications are expected to replicate or interpret.

Thus, we showed through our experiments that complex behaviors such as unfamiliar CSI prediction are inherently highly noisy and user-dependent, which makes a direct comparison of behaviors, as done in the more standard SDP paradigm, noisy and unreliable, which can be ameliorated by ISDP.

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

# A Appendix

## A.1 Prompts - SDP

---

**Prompt - Socio Demographic Prompting**

AI Rules
- Output response in JSON format
- Do not output any extra text.
- Do not wrap the outputs in JSON or Python markers
- JSON keys and values in double-quotes

You are a cultural mediator who understands all cultures across the world. As a mediator, your job is to identify and translate culturally exotic concepts from texts from an unknown source culture to my culture. I am a well-educated {genre} lover who grew up in {article_urban} urban {country}, which defines my culture. I came across a review of the book '{book}' by {author}, which belongs to the {book_genre} genre. Given my cultural background, perform the following tasks:

Task 1: Identify all culture-specific items (CSIs) from the review text that I might find hard to understand due to my cultural background. CSIs are textual spans denoting concepts and items uncommon and not prevalent in my culture, making them difficult to understand.

Task 2: For each CSI, identify its category from one of the following seven categories:
1. Ecology: Geographical features, flora, fauna, weather conditions, etc.
2. Material: Objects, artifacts, and products specific to a culture, such as food, clothing, houses, and towns.
3. Social: Hierarchies, practices, and rituals specific to a culture.
4. Customs: Political, social, legal, religious, and artistic organizations and practices. Customs, activities, procedures, and concepts.
5. Habits: Gestures, non-verbal communication methods, and everyday habits unique to a culture.
6. Linguistic: Terms unique to a specific language or dialect, including metaphors, idioms, proverbs, humor, sarcasm, slang, and colloquialisms.
7. Other: Anything not belonging to the above six categories.

Task 3: For each CSI, identify its familiarity from one of the following four levels:
1. Familiar: Most people from my culture know and relate to the concept as intended.
2. Somewhat familiar: Only some people from my culture know and relate to the concept as intended.
3. Unfamiliar: Most people from my culture do not know or relate to the concept.
4. Ambiguous: Most people from my culture know the concept, but its interpretation is varied or conflicting.

Task 4: For each CSI, identify its impact on the readability and understandability of the main point of the entire review text from one of the following three levels:
1. High: Greatly hinders the readability and comprehension of the review, making it difficult to convey its main points effectively.
2. Medium: It somewhat affects the readability and comprehension of the review, leading to only partial conveyance of its content.
3. Low: The review text's readability and comprehension will remain unaffected.

Task 5: Within 50 words, detail your reason for highlighting the span as CSI in Task 1 by correlating it with my background.

Task 6: Explain each CSI span within 20 words to make it more understandable to me. Provide facts, examples, equivalences, analogies, etc, if needed.

Task 7: Reformulate the entire text to make it more understandable to me. Keep the length similar to the original review text.

Format your response as a valid Python dictionary formatted as: {'spans': [List of Python dictionaries where each dictionary item is formatted as: {'CSI': <task 1: copy the CSI span from text>, 'category': <task 2: CSI category name>, 'familiarity': <task 3: familiarity level name>, 'impact': <task 4: impact level name>, 'reason': <task 5: reason within 50 words>, 'explanation': <task 6: explain the span within 20 words>]}, 'reformulation': <task 7: reformulate entire review text>}. Respond with {'spans': 'None'} if you think I will not find anything difficult to understand.

Text: {review_text}

---

## A.2 Prompts - ISDP

---

**Prompt - Inverse socio-demographic prompting (Review Level/User+Review Level)**

AI Rules
- Output response strictly in JSON format.
- Do not output any extra text or explanations outside the JSON.
- Do not wrap the outputs in JSON or Python markers.
- JSON keys and values should be enclosed in double quotes.

You are a cultural mediator who understands all cultures across the world. As a mediator, your job is to identify the cultural background of the users. The user came across a review of the book {book} by {author}, which belongs to the {book_genre} genre. Culture-specific items (CSIs) are textual spans denoting concepts and items uncommon and not prevalent in a culture, making them difficult to understand for an individual from a given culture. Given the review text and all the CSI spans that the users found hard to understand due to their cultural background, perform the following task.

Task 1: Based on the CSI spans that the users found difficult to understand due to their cultural background, provided as a tuple of (CSI span, number of users who found the span difficult to understand), identify the users' likely country of origin. Rank the countries in decreasing order of how unfamiliar the CSI spans are in that country. The possible countries are India, USA, and Mexico. The country ranked highest is the one where the CSIs are least common (most unfamiliar), and the lowest-ranked country is where the CSIs are most common (least unfamiliar).

Task 2: Explain within 20 words the reason behind the ranking of the countries in Task 1.

Format your response as a valid Python dictionary formatted as: {'country': <task 1: dictionary of all possible countries with their rankings formatted as {'country': ranking}>, 'reason': <task 2: reason within 20 words>}

Review Text: {review_text}

CSIs: $\{(\text{span}_1, n_1), (\text{span}_2, n_2), \ldots, (\text{span}_N, n_N)\}$

---

---

**Prompt - Inverse socio-demographic prompting (User Level)**

AI Rules
- Output response strictly in JSON format.
- Do not output any extra text or explanations outside the JSON.
- Do not wrap the outputs in JSON or Python markers.
- JSON keys and values should be enclosed in double quotes.

You are a cultural mediator who understands all cultures across the world. As a mediator, your job is to identify the cultural background of a user. Culture-specific items (CSIs) are textual spans denoting concepts and items uncommon and not prevalent in a culture, making them difficult to understand for an individual from a given culture. The user came across reviews of books. Given the review text and all the CSI spans that the user found difficult to understand due to their cultural background, perform the following task.

Task 1: Based on the CSI spans that the users found difficult to understand due to their cultural background, provided as a tuple of (CSI span, number of users who found the span difficult to understand), identify the users' likely country of origin.. Rank the countries in decreasing order of how unfamiliar the CSI spans are in that country. The possible countries are India, USA, and Mexico. The country ranked highest is the one where the CSIs are least common (most unfamiliar), and the lowest-ranked country is where the CSIs are most common (least unfamiliar).

Task 2: Explain within 20 words the reason behind the ranking of the countries in Task 1.

Format your response as a valid Python dictionary formatted as: {'country': <task 1: dictionary of all possible countries with their rankings formatted as {'country': ranking}>, 'reason': <task 2: reason within 20 words>}

Given below is the review of the book {book} by {author}, which belongs to the {book_genre} genre. Review Text1: {review_text}
CSIs: $\{(\text{span}_1, n_1), (\text{span}_2, n_2), \ldots, (\text{span}_N, n_N)\}$

Given below is the review of the book {book} by {author}, which belongs to the {book_genre} genre. Review Text: {review_text}
CSIs: $\{(\text{span}_1, n_1), (\text{span}_2, n_2), \ldots, (\text{span}_N, n_N)\}$
.
.
.
.
Given below is the review of the book {book} by {author}, which belongs to the {book_genre} genre. Review Text: {review_text}
CSIs: $\{(\text{span}_1, n_1), (\text{span}_2, n_2), \ldots, (\text{span}_N, n_N)\}$

---

## A.3 Results: Statistical Significance

| span_generator | discriminator1 | discriminator2 | t_statistic | p_value |
|---|---|---|---|---|
| aya | llama | aya | -5.326117 | 0.003124 |
| aya | llama | gemma | -1.347977 | 0.235514 |
| aya | llama | gpt4o | -0.450854 | 0.670974 |
| aya | aya | gemma | 0.227204 | 0.829263 |
| aya | aya | gpt4o | 1.572355 | 0.176675 |
| aya | gemma | gpt4o | 3.152963 | 0.025294 |
| gemma | llama | aya | -0.740100 | 0.492505 |
| gemma | llama | gemma | 0.413237 | 0.696558 |
| gemma | llama | gpt4o | 0.467127 | 0.660058 |
| gemma | aya | gemma | 0.769376 | 0.476424 |
| gemma | aya | gpt4o | 0.912667 | 0.403278 |
| gemma | gemma | gpt4o | -0.029062 | 0.977939 |
| gpt4o | llama | aya | 1.083931 | 0.327873 |
| gpt4o | llama | gemma | -1.355720 | 0.233205 |
| gpt4o | llama | gpt4o | 4.574447 | 0.005978 |
| gpt4o | aya | gemma | -2.422944 | 0.059899 |
| gpt4o | aya | gpt4o | 1.034381 | 0.348370 |
| gpt4o | gemma | gpt4o | 8.260590 | 0.000424 |
| user | llama | aya | 0.354713 | 0.737275 |
| user | llama | gemma | 0.343953 | 0.744874 |
| user | llama | gpt4o | -0.405212 | 0.702077 |
| user | aya | gemma | -0.251829 | 0.811197 |
| user | aya | gpt4o | -1.647885 | 0.160292 |
| user | gemma | gpt4o | -1.140894 | 0.305595 |
| llama | llama | aya | -3.120119 | 0.026249 |
| llama | llama | gemma | -0.835801 | 0.441365 |
| llama | llama | gpt4o | -1.177835 | 0.291865 |
| llama | aya | gemma | 4.332499 | 0.007482 |
| llama | aya | gpt4o | 1.412226 | 0.216988 |
| llama | gemma | gpt4o | -0.618236 | 0.563491 |

Table 2: Paired t-test results for review level experiments. Rows colored orange show that they are statistically significant (p_value < 0.05).

| span_generator | discriminator1 | discriminator2 | t_statistic | p_value |
|---|---|---|---|---|
| aya | llama | aya | -0.512273 | 0.630278 |
| aya | llama | gemma | 0.355522 | 0.736705 |
| aya | llama | gpt4o | 1.397820 | 0.221018 |
| aya | aya | gemma | 0.648904 | 0.545023 |
| aya | aya | gpt4o | 1.443309 | 0.208528 |
| aya | gemma | gpt4o | 1.843266 | 0.124623 |
| gemma | llama | aya | -0.569640 | 0.593560 |
| gemma | llama | gemma | -1.727200 | 0.144713 |
| gemma | llama | gpt4o | 1.799847 | 0.131783 |
| gemma | aya | gemma | -0.486936 | 0.646898 |
| gemma | aya | gpt4o | 2.736133 | 0.040980 |
| gemma | gemma | gpt4o | 6.344467 | 0.001436 |
| gpt4o | llama | aya | 3.633479 | 0.015006 |
| gpt4o | llama | gemma | -0.968320 | 0.377358 |
| gpt4o | llama | gpt4o | 8.703857 | 0.000331 |
| gpt4o | aya | gemma | -4.606414 | 0.005806 |
| gpt4o | aya | gpt4o | 5.604685 | 0.002499 |
| gpt4o | gemma | gpt4o | 12.269834 | 0.000064 |
| user | llama | aya | -1.301253 | 0.249904 |
| user | llama | gemma | -4.421119 | 0.006885 |
| user | llama | gpt4o | -3.433467 | 0.018566 |
| user | aya | gemma | -0.779455 | 0.470977 |
| user | aya | gpt4o | -1.842872 | 0.124686 |
| user | gemma | gpt4o | -2.003821 | 0.101443 |
| llama | llama | aya | 2.007264 | 0.100998 |
| llama | llama | gemma | 0.598369 | 0.575666 |
| llama | llama | gpt4o | 6.860223 | 0.001006 |
| llama | aya | gemma | -1.150194 | 0.302086 |
| llama | aya | gpt4o | 4.116861 | 0.009202 |
| llama | gemma | gpt4o | 7.476294 | 0.000676 |

Table 3: Paired t-test results for user+review level experiments. Rows colored orange show that they are statistically significant (p_value $< 0.05$).

| span_generator | discriminator1 | discriminator2 | t_statistic | p_value |
|---|---|---|---|---|
| aya | llama | aya | -0.466986 | 0.660152 |
| aya | llama | gemma | -0.800898 | 0.459539 |
| aya | llama | gpt4o | -0.755307 | 0.484104 |
| aya | aya | gemma | -0.679548 | 0.526970 |
| aya | aya | gpt4o | -0.314223 | 0.766040 |
| aya | gemma | gpt4o | -0.063718 | 0.951664 |
| gemma | llama | aya | -0.036038 | 0.972647 |
| gemma | llama | gemma | 0.553196 | 0.603952 |
| gemma | llama | gpt4o | 1.631805 | 0.163649 |
| gemma | aya | gemma | 0.951767 | 0.384923 |
| gemma | aya | gpt4o | 2.026568 | 0.098541 |
| gemma | gemma | gpt4o | 2.157585 | 0.083441 |
| gpt4o | llama | aya | -2.018184 | 0.099601 |
| gpt4o | llama | gemma | -0.724634 | 0.501154 |
| gpt4o | llama | gpt4o | -2.326693 | 0.067489 |
| gpt4o | aya | gemma | 2.038030 | 0.097112 |
| gpt4o | aya | gpt4o | -1.258016 | 0.263937 |
| gpt4o | gemma | gpt4o | -3.157931 | 0.025154 |
| user | llama | aya | 0.379708 | 0.719756 |
| user | llama | gemma | -1.288302 | 0.254034 |
| user | llama | gpt4o | -1.117324 | 0.314649 |
| user | aya | gemma | -1.075421 | 0.331318 |
| user | aya | gpt4o | -1.512893 | 0.190718 |
| user | gemma | gpt4o | 0.382935 | 0.717507 |
| llama | llama | aya | 0.117893 | 0.910742 |
| llama | llama | gemma | 0.872386 | 0.422905 |
| llama | llama | gpt4o | 1.005745 | 0.360700 |
| llama | aya | gemma | 0.910761 | 0.404190 |
| llama | aya | gpt4o | 0.502238 | 0.636832 |
| llama | gemma | gpt4o | -0.395285 | 0.708934 |

Table 4: Paired t-test results for user level experiments. Rows colored orange show that they are statistically significant ($p\_value < 0.05$).

## B   Dataset standardization

The dataset contains human annotations of 57 English reviews of books originating from India, UAE, and the USA. The annotations are performed by 50 users, where 8 are from India, 22 are from Mexico, and 20 are from the USA. Given a book review, each user identifies text spans of varying lengths that they found difficult to understand.

**Span standardization:**    Since users can mark any contiguous text spans as difficult to understand, there is a high degree of variability in the annotations. To handle this, the original dataset (Saha et al., 2025) semantically clustered the spans using sentence transformers[3] and filtered out poor-quality annotations before conducting quantitative and qualitative analysis. However, this approach has several limitations: **(i) User-User Mismatch:** The lengths of user annotations vary where one user might identify multiple CSIs as a single contiguous span, whereas others might segment the same span into multiple non-contiguous spans. For example, the span *from the Beats in On The Road to Ken Kesey's Merry Pranksters* refers to two influential countercultural movements in American literature and history represented by *the Beats in On The Road* and *Ken Kesey's Merry Pranksters*. Someone unfamiliar with either of them might mark the entire span as difficult to understand, while others might partially mark the spans, signifying familiarity with either *the Beat generation* or *the pranksters*. Some might non-contiguously highlight both spans to indicate unfamiliarity with both of them. Such fine-level distinctions are lost in sentence transformers-based semantic clustering. **(ii) User-Model Mismatch:** User-annotated spans were generally longer than those generated by the model, primarily due to differences in word boundary recognition. Users often group multiple CSIs into a single span, whereas the model-generated responses tend to break them down into more atomic CSIs. **(iii) Model-Model Mismatch:** Unlike Saha et al. (2025), since we also evaluate multiple models here, there might be scenarios where the model-highlighted spans do not have a 1:1 match, similar to the user-user mismatches. To alleviate these issues, we collected all the CSI spans annotated by users and models for a given review text and manually standardized the spans for each review text, ensuring consistent discourse segmentation.

---

*Before Standardization:*

**User:** John Muir, Muir woods, Stickeen, The Moral Equivalent of War by William James
**Model 1:** John Muir? Sure, Muir woods,
**Model 2:** John Muir
**Overlap scores. Model 1:** 1.0 **Model 2:** 1.0

*After Standardization:*

**User:** John Muir#Muir woods#Stickeen#The Moral Equivalent of War#William James
**Model 1:** John Muir#Muir woods
**Model 2:** John Muir
**Overlap scores. Model 1:** 0.4 **Model 2:** 0.2

---

Above is an illustrative example, which shows the overlap scores for two models before and after the standardization process. Before standardization, both models 1 and 2 attain a perfect score using sentence transformers with a similarity threshold of 0.5. However, after standardization, the user span is split into five unique spans depicting different CSIs. Using an exact match, we obtain an overlap score of 0.4 for Model 1 and 0.2 for Model 2.

The dataset contains 1,193 combinations of reviews and CSIs annotated by the users and generated by the models. Three Computer Science and Linguistics experts manually standardized all 1,193 spans in the dataset by converting them to their appropriate elementary discourse units (EDUs). Each annotator annotated 450 spans (avg 19 reviews per annotator), which were randomly sampled and had ~50 overlapping spans among them to facilitate calculating inter-annotator agreement (IAA) scores. The annotation guidelines for the standardization of spans are as follows.

- If a span contains multiple CSIs, split it into their elementary discourse units separated by a "#" symbol.

---

[3]sentence-transformers/all-MiniLM-L6-v2

- If a span contains only part of a named entity (such as a book title, album title, or proper noun), the span should be expanded to include the full entity.
- Correct any grammatical errors and formatting inconsistencies, wherever necessary.

Since each annotation either involved correcting the errors in the span or splitting a span into multiple spans, we use a mean-based IAA metric to calculate agreement between the three expert annotators. We assign a weighted score to the agreements and disagreements while calculating IAA and assign a perfect score if the two annotators agree on an annotation and a score of 0.7 otherwise. After the first round of annotation, we obtain a mean agreement of 0.967. The disagreements were discussed and resolved by an additional round of annotation. We observe a decrease in the average number of words in user spans from 6.31 to 3.30, and from 5.22 to 3.36 in the model spans, indicating better consistency. Post standardization, the total review text and span combinations were 922 (322 from users, 600 from models), compared to 1,193 (365 from users, 828 from models). Overall, the dataset contains 671 unique spans across all review texts, compared to 1,122 spans. The standardization process ensures the reliability of any following span-based analysis, enabling more robust comparisons between humans and models.[4]

---

[4]Contact authors for the dataset.

