# OpenReview forum: "All Norms and No Nuance Make LLMs Dull Cultural Simulators"
_colmweb.org/COLM/2025/Workshop/Social_Sim — Social Sim'25_

### Official Review · Reviewer_t5vo · 2025-07-04
**All Norms and No Nuance Make LLMs Dull Cultural Simulators Review**

**Rating:** 8
**Overall Assessment:** 4
**Confidence:** 3

**Review:**

I find this paper very important as well as related to the workshop topics, and believe that this paper is of interest to the community. The presentation quality is good and clear, with good results. However, the work is limited by its narrow scope (single dataset, three countries) and also on when one should use their framework.

**Comments Suggestions And Typos:**

Overall, I think adding a more detailed discussion on when and how practitioners should use ISDP vs. SDP in real applications could improve the quality of the paper and would be beneficial.

**Paper Summary:**

This paper introduces Inverse Socio-Demographic Prompting (ISDP), a method to assess cultural biases in LLMs by predicting users' cultural backgrounds from their behaviors, rather than generating behaviors from demographic proxies (SDP). The authors evaluate four LLMs (Aya-23, Gemma-2, GPT-4o, Llama-3.1) on the Goodreads-CSI dataset, which captures cross-cultural non-understandability in book reviews from users in three countries. They find that ISDP is more robust than SDP for measuring cultural alignment, with GPT-4o outperforming other models in handling nuanced, non-stereotypical behaviors. However, LLMs generally struggle to simulate real user behavior accurately, often generating stereotypical outputs due to maximum likelihood decoding.

**Relevance:**

5

**Summary Of Strengths:**

1. The most important contribution of this paper is its idea of ISDP, which is both novel and important.
2. The selection of the dataset, Goodreads-CSI dataset, is a wise choice. (While the diversity of countries and cultures inside the dataset is limited, the data itself is a reasonable choice.
3. The use of multiple metrics (MRR, IAA) and statistical significance testing strengthens the validity of findings.

**Summary Of Weaknesses:**

1. While I find the idea interesting, the evaluation is restricted to only one dataset (Goodreads-CSI) with a few countries, limiting generalizability. The findings may not hold across different cultural contexts or tasks, but I understand the limitations of the academic budget.
2. While ISDP is proposed as a "robust alternative," the paper doesn't provide clear guidance on when and how practitioners should use ISDP versus SDP in real applications.
3. The study focuses primarily on country as a cultural proxy, but doesn't adequately explore other demographic factors or intersectionality.

---

### Official Review · Reviewer_Vuim · 2025-07-08
**A good workshop submission with interesting angles.**

**Rating:** 6
**Overall Assessment:** 3
**Confidence:** 4

**Review:**

Quality - the paper is *good* in quality comparing to standard workshop submissions with extensive analysis and experiments.

Clarity - the clarity is *fair*, see comments in "Summary Of Weaknesses".

Originality - the paper has *good* originality, framing demographic predictions based on text as an alternative of mapping behaviours to culture.

Significance - the significance of the work is *fair*, see "Summary Of Strengths".

**Comments Suggestions And Typos:**

-  The presentation is sometimes dense and inconsistent, especially in the introduction and in terms of the story of the paper. I think the paper could greatly benefit from a concrete definition of cultural “behaviour” or cultural norms in the paper, or perhaps a reframing. (see weaknesses)
- If possible, include more cultures/data or at least discuss plans and challenges for generalizing beyond the Goodread-CSI.

**Paper Summary:**

In this paper, the authors propose “Inverse Socio-Demographic Prompting” (ISDP), which asks models to infer cultural background from user-annotated behaviours, instead of the common “socio-demographic prompting” (SDP) for probing Large Language Models (LLMs) for cultural alignment. The authors argue that SDP’s sensitivity to prompt phrasing leads to unreliable measurements. The paper compares SDP and ISDP on the Goodreads-CSI dataset across four models, three levels of behaviour aggregation, and multiple behaviour sources (user vs. model‐generated) demonstrate that (1) ISDP yields more reliable discrimination than SDP, (2) models often generate stereotypical behaviours under SDP that deviate from real users, and (3) GPT-4o exhibits superior nuance at the individual level.  Through the analysis the paper provide interesting insights on understanding the limitations of user behaviour simulations.

**Relevance:**

4

**Summary Of Strengths:**

- The paper focuses on a timely and interesting problem.
- The angle of  “Inverse Socio-Demographic Prompting” (ISDP) is an interesting approach and provides fresh perspectives.
- While only experimented with one dataset, the experimental designs with aggregations across 3 levels and 4 different LLM shows diligent efforts.

**Summary Of Weaknesses:**

- There are several conceptual weakness with the proposed work:
  - The authors’ definition of cultural “behaviour” would benefit from greater clarity and tighter linkage between concept and experiment. The authors adopt an umbrella definition using “norms” “value” etc. and referring to recent surveys in Cultural NLP. In practice, “behaviour” in the experiment is operationalized as highlighting confusing spans (CSIs) in text from the Goodread-CSI dataset. The connection between marking a difficult phrase and embodying genuine cultural norms or values is not fully motivated.
  - It remains uncertain whether model performance reflects a genuine understanding of cultural norms/values or the exploitation of linguistic artifacts such as uncommon vocabulary, syntactic structures, or punctuation patterns. Without appropriate controls, the results might be a reflection of surface-level pattern recognition. Although this could be referred to as cultural “behaviour”, but again, imho requires concrete and precise definition of the term, and is consistent with the experiments.
- Currently, it is also unclear whether commercial LLMs have been trained or fine-tuned on proprietary sources like Goodreads, which could introduce data leakage. If such sources are present, accurate identifications may reflect memorization or indirect cues from leaked content.

---

### Meta-Review · Program_Chairs · 2025-07-24

**Recommendation:** Accept

**Metareview:**

--